# Dietary patterns and associated factors among pregnant women in Ibadan, Nigeria: Evidence from Ibadan pregnancy cohort study

Ikeola A. Adeoye[1,2]*, Akinkunmi P. Okekunle[3]

**1** Department of Epidemiology and Medical Statistics, College of Medicine, University of Ibadan, Ibadan, Nigeria, **2** Consortium for Advanced Research in Africa (CARTA), Nairobi, Kenya, **3** Research Institute of Human Ecology, Seoul National University, Seoul, Korea

* adeoyeikeola@yahoo.com

## Abstract

### Background

Maternal nutrition is vital for an optimal intrauterine environment, foetal development, birth weight, pregnancy and neonatal outcomes. We assessed the maternal dietary patterns using a data-driven technique and the associated sociodemographic factors among pregnant women in Ibadan, Nigeria.

### Methodology

Dietary assessment was performed during the enrolment of participants for the Ibadan Pregnancy Cohort Study, a prospective cohort study, conducted among 1745 pregnant women enrolled early in pregnancy (≤ 20 weeks) at four comprehensive obstetric facilities within the Ibadan metropolis. A qualitative food frequency questionnaire was used to assess the pregnant population's intake of food and drinks three months prior to their enrollment. We determined dietary patterns by applying principal component analysis with a varimax rotation. Multivariate analysis was used to investigate the association between sociodemographic factors and dietary patterns at 5% statistical significance.

### Results

Mean age and gestational age at enrolment were 29.8 (± 5.3) years and 16.4 (±4.2) weeks, respectively. White rice was the most frequently consumed meal [794 (45.5%) daily, 898 (51.4%)] weekly in our study population. Five major dietary patterns were identified, and they accounted for 28.8% of the total variation: "protein-rich diet with non-alcoholic beverages" (15.6%); "fruits" (4.1%); "typical diet with alcohol" (3.8%); "legumes" (2.8%), "refined grains" (2.6%). Maternal education and income were inversely associated with the consumption of a "protein-rich diet with non-alcoholic beverages", "typical diet with alcohol", and "legumes" in a dose-response fashion. Also, employed women had a higher mean intake of fruits [adjusted β: 0.33 (0.02; 0.65) p = 0.040] compared with women without employment.

**Data Availability Statement:** Availability of data and materials: The datasets generated and/or analysed during the current study are not publicly available because they contain potentially

identifying and confidential information but are available from the UI/UCH Ethics Committee (uiuchec@gmail.com) on reasonable request if it meets the criteria for accessing confidential data.

**Funding:** This research was supported by the Consortium for Advanced Research Training in Africa (CARTA). CARTA is jointly led by the African Population and Health Research Center and the University of the Witwatersrand and funded by the Carnegie Corporation of New York (Grant No. G-19-57145), Sida (Grant No:54100113), Uppsala Monitoring Center, Norwegian Agency for Development Cooperation (Norad), and by the Wellcome Trust [reference no. 107768/Z/15/Z] and the UK Foreign, Commonwealth & Development Office, with support from the Developing Excellence in Leadership, Training and Science in Africa (DELTAS Africa) programme. Ikeola Adeoye is a CARTA PhD fellow. The statements made and views expressed are solely the responsibility of the Fellow. The funders had no role in study design, data collection and analysis, decision to publish, or preparation of the manuscript. For the purpose of open access, the author has applied a CC BY public copyright licence to any Author Accepted Manuscript version arising from this submission.

**Competing interests:** The authors declare that they have no completing interest.

## Conclusions and recommendation

We described five dietary patterns of pregnant women using a data-driven technique, principal component analysis, in Nigeria. We also identified factors influencing maternal dietary patterns, which can inform public health interventions, especially behavioural change communication during antenatal care.

## Introduction

Nutrition and pregnancy are closely linked because maternal nutrition influences the intra-uterine environment [1, 2]. Maternal nutrition is also an important modifiable determinant of foetal development, birth weight, pregnancy and neonatal outcomes [3]. For example, micronutrient deficiencies predominant among pregnant women in low and middle-income countries (LMICs) are risk factors for iron deficiency anaemia, low birth weight, intrauterine growth restriction, and small gestational age [4–6]. However, excess energy intake is a risk factor for obesity, excessive gestation weight gain, gestational diabetes mellitus (GDM), and macrosomia [7, 8] which are emerging public health concerns in LMICs.

Determining the overall significance of food and dietary consumption in a population can be complex, as a single nutrient approach (such as iron, iodine, and folate deficiencies) is limited because nutrients are not consumed singly in diets but rather in the company of several foods/nutrients over time. The overall food and dietary pattern assessment is a suitable methodology for summarising overall food and dietary consumption in populations [9]. Dietary patterns assessment is an objective evaluation of a population's overall food and dietary exposure and is often used in determining the diet-disease association in nutrition epidemiology [10]. It utilises the amount, type, frequency or combination of different foods and beverages typically consumed over time [11] to provide a broader picture of the whole food and nutrient consumption [9, 10]. The dietary pattern approach has increasingly gained popularity in explaining the relationship between habitual diet and chronic disease risk. For instance, increasing quartiles of westernized diet were associated with the risk of coronary artery disease in the United States [12]. Although dietary patterns have received scant attention among pregnant women in LMICs such as Nigeria, certain foods have been associated with lowering the risk of some pregnancy complications. For example, diets high in whole grains, fish, fruits and vegetables have reportedly lowered the risk of gestational diabetes mellitus [13] and gestational hypertension [14].

However, the significance of overall dietary exposure in maternal and neonatal outcomes in LMICs, including in Nigeria, has not been thoroughly investigated. Also, the sociodemographic factors associated with maternal dietary patterns have not been identified in Nigeria. Identifying the overall food consumption and its associated factors among pregnant women is crucial for designing and implementing appropriate nutritional education, counselling and public health interventions for improving maternal and neonatal outcomes, particularly in the light of the escalating burden of maternal obesity in LMICs. Therefore, this study examined derived dietary patterns and the associated factors among pregnant women in Ibadan using the Ibadan Pregnancy Cohort Study.

## Materials and methods

### Study design, setting and population

The Ibadan Pregnancy Cohort Study (IbPCS) was a multicentre hospital-based study among women and their offspring aimed at assessing the association of maternal obesity and lifestyle

factors with glycaemic control, gestational weight gain, pregnancy and postpartum outcomes in Ibadan, Nigeria. The study started in April 2018, and the baseline recruitment was completed in March 2019. Details of its methodology have been reported elsewhere [15]. It was a prospective cohort study that recruited 1745 pregnant women in early pregnancy (≤20 weeks) from the four health facilities within the Ibadan metropolis. The study was facility-based and conducted at four hospitals which are major maternal health care services providers and referral centres for comprehensive essential obstetric care within the Ibadan metropolis. These facilities are University College Hospital, Adeoyo Maternity Teaching Hospital, Jericho Specialist Hospital, and Saint Mary Catholic Hospital, Oluyoro, Ibadan.

## Data collection procedures

Data were collected using pretested, interviewer-administered questionnaires and structured proforma at booking, third trimester, and delivery. Trained personnel conducted in-person interviews and physical examinations (using standard instruments) to assess information on sociodemographic and lifestyle characteristics and dietary information from respondents at baseline after due informed consent for the study. Sociodemographic information assessed were age (in years), Yoruba ethnicity/ancestry (no or yes), level of education completed (at least primary, secondary or tertiary), average monthly income in naira—N (<N20,000; N20,000—N99,999 or ≥ N100,000), employment status (no, yes), religion (Christianity or Islam) and marital status (single or married). Also, respondents reported the number of birth experiences they have had prior to the current pregnancy and were classified as; nulliparous, 1–3 or ≥4.

**Dietary assessment.** Participants provided information on foods and drinks consumed in the last three months using an interviewer-administered qualitative food frequency questionnaire (FFQ). The FFQ was designed from a sampling frame of foods and drinks reported by fifty randomly selected women of reproductive age using a 24-hour dietary recall. The FFQ was made of 67 food and drinks classified into ten food groups' cereals', 'starchy roots and tubers', 'legumes', 'meat, fish and poultry products', 'fruits', 'vegetables', 'milk', 'sugar-sweetened beverages and drinks', 'alcohol' and 'pastries'. Details of the food and drink items in the FFQ and how they are classified into food groups are presented in Table 1. For each food or drink, participants reported the frequency of food consumption as follows: once daily, more than once daily (i.e. 2–3 times daily): once weekly, more than once weekly (i.e. 2–3 times weekly): once monthly, more than once monthly (i.e. 2–3 times monthly). The consumption frequency was harmonised into daily, weekly, monthly and rarely and transformed into the frequency of daily consumption.

**Dietary pattern analysis.** The dietary pattern was derived using daily frequency consumption of 67 food items, with a reliability coefficient of 91.0%. Principal Component Analysis (PCA) [16–18], was applied to the correlation matrix of the daily consumption frequency of 67 food items. The factor loadings of foods and drinks were estimated using an orthogonal varimax rotated transformation for interpretability and extraction of uncorrelated components/ factors. Five factors were retained based on an eigenvalue, a scree plot and interpretability. Factor loadings were calculated for each food item, and factors were interpreted as dietary patterns. Food or drink items having an absolute loading of ≥0.20 were retained in each dietary pattern. Factor scores of respondents in each dietary pattern were estimated, with higher factor scores typifying a level of closeness of the foods/drinks to the dietary patterns and vice versa. In order to determine the level of adherence to dietary patterns, respondents' factor scores in each dietary pattern were ranked and stratified into 'low' if the respondent factor score in a dietary pattern falls within the 50th percentile of the factor score distribution in this sample, otherwise 'high' where the factor score is > 50th percentile.

**Table 1. Food items and food groups used in the dietary patterns.**

| Food Groups | Food Items |
|---|---|
| Cereals and Products: | White rice, Jollof rice, Fried rice, Ofada rice, Spaghetti, White Bread, Wheat Bread, Wheat Semovita, Pap, Cornflakes, Oats, Golden Morn, Wheaterbix/All bran/Fruit fibre. |
| Starch Roots and Tubers | *Eba*, *Amala*, Pounded yam, *Fufu*, Yam Porridge, Pando yam/, Boiled Yam, Boiled Potato |
| Legumes | Stewed beans *(Ewa riro)*, *Moinmoin*, *Ekuru*, *Gbegiri*, Cowpea–*feregede* |
| Meat and Fish | Red meat–beef, Pork, Internal organs/offals–*shaki*, liver, lungs, Snails, Fish, Poultry–Chicken or turkey, Eggs |
| Fruits | Pawpaw, Watermelon, Pineapple, Apples, Tangerine/tangelo, Cucumber, Avocado pear, English Pear, Oranges, Carrots, Mangoes, Banana, *Agbalumo* (cherry) |
| Vegetables | Plain Vegetable soup–Okro, *Ewedu*, *Efo Riro*, *Egusi Soup*, *Ogbonno* / Groundnut /*Afan/Oha Soup*, Garden egg, Corn, Cabbage |
| Milk | Cream Milk, Skimmed or low-fat milk, Soya Milk, Kunu |
| Sugar-Sweetened Drinks | Soft drinks, Malt drinks, Fruit juice, Beverage–Milo, Bournvita, Tea, Coffee, Yoghurt. |
| Alcohol | Beer, Palm wine, Whisky/dry Gin |
| Pastries | Cake, puff-puff, doughnut, Buns, chinchin |

Number of items in the scale: 67.

Scale reliability coefficient: 0.9101.

## Statistical analysis

Univariate analysis was performed in which categorical and continuous data were presented using percentages and mean (standard deviation). Also, by food groups the frequencies of commonly consumed food items were presented in composite bar graphs. The bivariate analysis examined the five dietary patterns' sociodemographic and lifestyle characteristics between low and high levels. Linear regression was used to estimate the beta (β) coefficient and 95% confidence intervals (CI) of sociodemographic and lifestyle factors and scores for each dietary pattern. Furthermore, only statistically significant variables in the unadjusted linear regression models were included in the final linear regression models to estimate the adjusted β coefficient and 95% CI of factors associated with each dietary pattern in this sample. All statistical analyses were carried out at a two-sided $P < 0.05$.

## Ethical consideration

The ethical approval for this study was obtained from the University of Ibadan/University College Hospital (UI/UCH) Institutional Review Board (UI/EC/15/0060) and Oyo State Ministry of Health Ethical Committee (AD/13/479/710). Verbal and written informed consent was obtained from the respondents before recruitment into the study. The study protocol and conduct adhered to the principles laid down in the Declaration of Helsinki.

## Results

### Characteristics of study participants (Table 2) and their food consumption pattern (Fig 1)

The characteristics of the Ibadan pregnancy cohort are presented in Table 2. The mean age was 29.8 (± 5.3) years, and the mean gestational age at enrolment was 16.4 (±4.2) weeks. Also, the majority of the women were within 20–39 years 1648 (94.4%), married 1643 (94.1%), employed 1557 (89.2%), and of Yoruba ancestry 1565 (89.8%). About a third, 583 (33.4%)

**Table 2. Characteristics of the study participants.**

| Characteristics | N (frequency) | Percentage (n/N) |
|---|---|---|
| **Age group** | | |
| < 20 | 28 | 1.6 |
| 20–29 | 844 | 48.3 |
| 30–39 | 804 | 46.1 |
| ≥ 40 years | 70 | 4.0 |
| **Mean age (years)** | **29.8 (±5.3)** | |
| **Mean gestational age (weeks)** | **16.4 (±4.6)** | |
| **Parity** | | |
| Nulliparous | 761 | 43.8 |
| 2–4 | 882 | 50.7 |
| ≥ 5 | 95 | 5.5 |
| **Marital Status** | | |
| Single | 102 | 5.8 |
| Married | 1644 | 94.2 |
| **Level of Education** | | |
| ≤ Primary | 49 | 2.8 |
| Secondary | 504 | 28.9 |
| Tertiary | 1189 | 68.3 |
| **Occupation** | | |
| Employed | 1557 | 89.2 |
| Unemployed | 189 | 10.5 |
| **Religion** | | |
| Christianity | 1011 | 58.1 |
| Islam | 730 | 41.9 |
| **Ethnicity** | | |
| Yorubas | 1565 | 89.8 |
| Non-Yorubas | 177 | 10.2 |
| **Income per month (Naira)** | | |
| <20,000 | 583 | 33.4 |
| 20,000–99,999 | 501 | 28.7 |
| ≥ 100,000 | 108 | 6.2 |

earned less than N20,000 per month (minimum wage), and 1011 (58.1%) reported being
Christians.

White rice was the most frequently consumed meal (Fig 1) among the study participants:
[794 (45.5%) daily, 898 (51.4%) weekly]. Legumes were consumed mostly on a weekly basis:
stewed beans (*Ewa riro*) [158 (9.1%) daily & 1200 (68.7%) weekly] Bean Pudding (*Moinmoin*)
[128 (7.3%) daily & 1176 (67.4) weekly] Beans cake *(Akara)* [158 (6.3%) daily & 1022 (58.5)
weekly]. The commonest sources of animal protein were red meat: [816 (46.7%) daily & 612
(35.1%)] weekly, fish: [817 (46.8%) daily & 696 (39.9%) weekly], and eggs [645 (36.9%) daily &
831 (47.6%) weekly]. Plain vegetable soups, such as Okro, *Ewedu, and Efo riro*, were the most
frequently consumed in the fruits and vegetable group.

## Dietary pattern of study participants (Table 3)

The factor loading matrix of the dietary patterns obtained from the PCA is presented in
Table 3. Five major dietary patterns, which accounted for 28.8% of the total variation, were

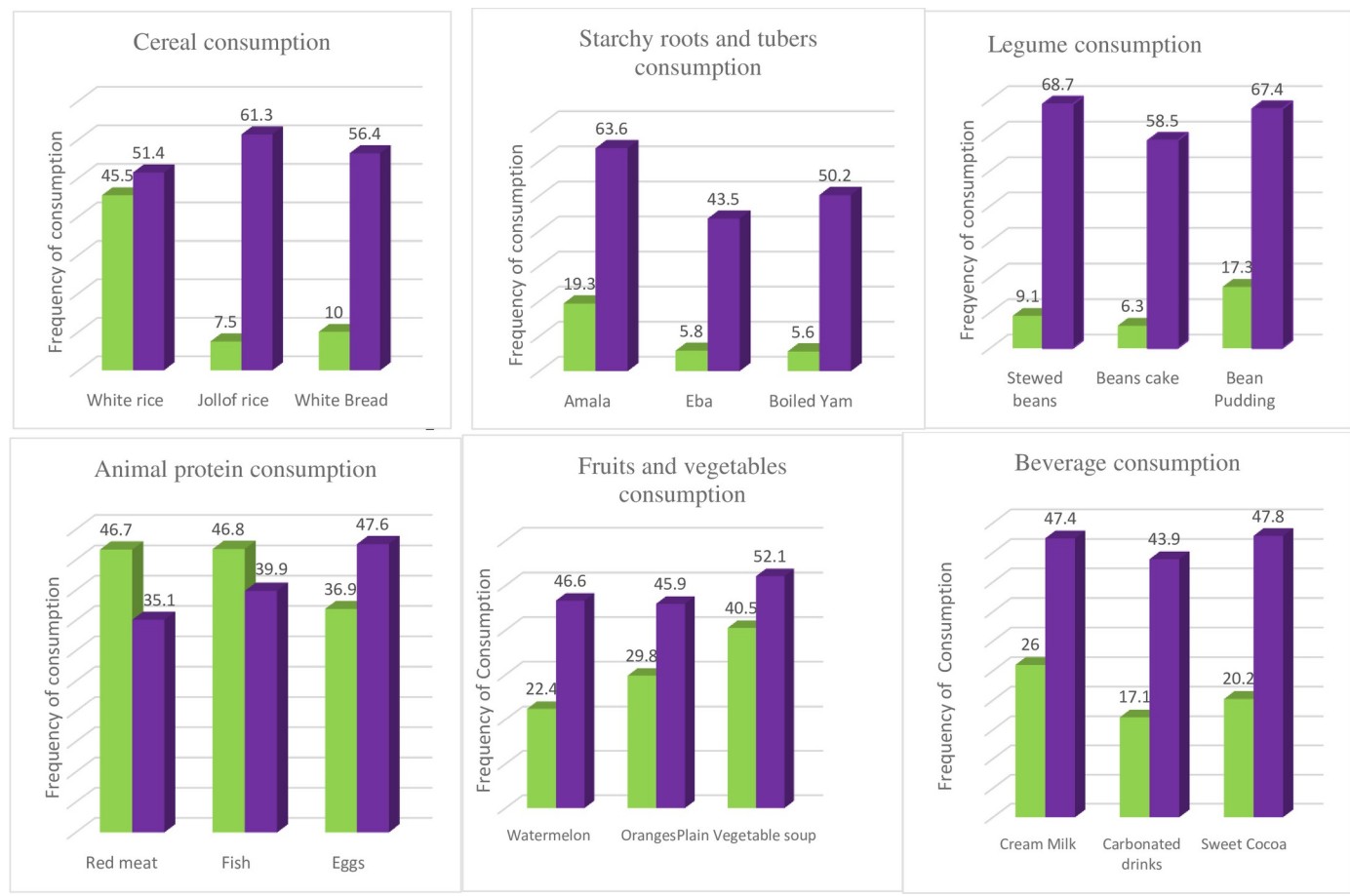

**Fig 1. Most commonly consumed food items by food groups in the Ibadan pregnancy cohort daily (green); weekly (purple).**

identified in the study population. The first pattern, "protein-rich diet with non-alcoholic beverages", accounted for the highest variance (15.6%) and was characterised by a high intake of red meat, fish, eggs, green vegetables, cream milk, and soft drinks, cocoa beverages, and pastries. The second pattern, "fruits", which accounted for 4.1% variance, was characterised by pawpaw, watermelon, pineapple, tangerine/tangelo, cucumber, avocado pear, carrots, mangoes and cherry. The "typical diet with alcohol" was the third pattern, characterised by *pando yam*, fermented cassava pudding *(fufu)*, pork, snail, soya milk, beer, and palm wine. The fourth pattern, "legumes", explained a 2.8% variance and was characterised by stewed beans, bean cake, beans pudding *(moinmoin)* bland beans pudding *(ekuru)*. The last pattern was termed "refined grains" and was characterised by high consumption of jollof rice, fried rice, *ofada rice*, corn flakes, oats and golden morn, explaining 2.6% of the total variance.

## Association between maternal sociodemographic characteristics and dietary patterns (Tables 4 and 5)

Table 4 displays the association between the sociodemographic factors and dietary patterns–a protein-rich diet with non-alcoholic beverages and fruits and a typical diet with alcohol, legumes, and refined grains among study participants. The protein-rich diet with non-alcoholic beverages, legumes pattern, and refined grains varied by education, income and religion. The consumption of a protein-rich diet with non-alcoholic beverages (primary "40.8%" versus

**Table 3. Factor loading matrix of 5 dietary patterns obtained by principal component analysis.**

| Food Items | Protein rich diet and non-alcoholic beverages | Fruits | Typical diet with alcohol | Legumes | Refined grains |
|---|---|---|---|---|---|
| Jollof Rice | - | - | - | - | 0.2815 |
| Fried Rice | - | - | - | - | 0.3537 |
| Ofada Rice | - | - | - | - | 0.2943 |
| Cornflakes | - | - | - | - | 0.3906 |
| Oats | - | - | - | - | 0.3184 |
| Golden Morn | - | - | - | - | 0.3031 |
| Fufu | - | - | 0.2044 | - | - |
| Pando Yam | - | - | 0.2346 | - | - |
| Stewed beans | - | - | - | 0.3104 | - |
| Bean Cake | - | - | - | 0.3762 | - |
| Moinmoin | - | - | - | 0.3590 | - |
| Ekuru | - | - | - | 0.3641 | - |
| Gbegiri | - | - | - | 0.2435 | - |
| Red Meat | 0.2432 | - | - | - | - |
| Pork | - | - | 0.2635 | - | - |
| Snail | - | - | 0.2317 | - | - |
| Fish | 0.2137 | - | - | - | - |
| Eggs | 0.2025 | - | - | - | - |
| Pawpaw | - | 0.3101 | - | - | - |
| Watermelon | - | 0.2918 | - | - | - |
| Pineapple | - | 0.3001 | - | - | - |
| Apple | - | 0.2887 | - | - | - |
| Tangerine | - | 0.2634 | - | - | - |
| Cucumber | - | 0.3107 | - | - | - |
| Avocado Pear | - | 0.2621 | - | - | - |
| English Pear | - | 0.2563 | - | - | - |
| Oranges | - | 0.2108 | - | - | - |
| Carrot | - | 0.2892 | - | - | - |
| Mangoes | - | 0.2612 | - | - | - |
| Agbalumo | - | 0.2567 | - | - | - |
| Plain vegetables | 0.2543 | - | - | - | - |
| Oil-based soups | - | - | 0.2337 | - | - |
| Cream milk | 0.3059 | - | - | - | - |
| Soya Milk | - | - | 0.2344 | - | - |
| Soft drinks | 0.2771 | - | - | - | - |
| Malt drinks | 0.2398 | - | - | - | - |
| Beverage | 0.3116 | - | - | - | - |
| Tea | 0.2244 | - | - | - | - |
| Beer | - | - | 0.2067 | - | - |
| Palm wine | - | - | 0.2358 | - | - 0.2388 |
| Pastries | 0.2079 | - | - | - | - |
| % Variance | 15.62 | 4.12 | 3.79 | 2.79 | 2.57 |
| %Cumulative variance | 28.88 | | | | |

Varimax rotated factor loadings $\geq$ 0.2 presented.

tertiary "30.3%") and legumes (primary "44.9%" versus tertiary "29.3%") was less among women with higher education compared with those with less education. On the other hand,

**Table 4. Association between participants' sociodemographic characteristics and dietary patterns.**

| | Protein-rich diet with non-alcoholic beverages | | | Fruits | | | Typical diet with alcohol | | | Legumes | | | Refined grains | | |
|---|---|---|---|---|---|---|---|---|---|---|---|---|---|---|---|
| | Low | High | p-value | Low | High | p-value | Low | High | p-value | Low | High | p-value | Low | High | p-value |
| **Age group** | | | | | | | | | | | | | | | |
| < 20 | 13(39.4) | 20 (60.6) | 0.172 | 17 (51.5) | 16 (48.5) | 0.641 | 19 (57.6) | 14 (42.4) | 0.741 | 19 (57.6) | 14 (42.4) | 0.551 | 15 (45.4) | 18 (54.6) | 0.134 |
| 20–29 | 401 (48.2) | 431 (51.8) | | 415 (49.9) | 417 (50.1) | | 421 (50.6) | 411 (49.4) | | 404 (48.6) | 428 (51.4) | | 396 (47.6) | 436 (52.4) | |
| 30–39 | 420 (51.7) | 392 (48.3) | | 412 (50.7) | 400 (49.3) | | 398 (49.0) | 414 (51.0) | | 417 (51.4) | 395 (48.7) | | 422 (52.0) | 390 (48.0) | |
| ≥ 40 years | 40 (57.4) | 29 (42.7) | | 29 (42.7) | 39 (57.4) | | 35 (51.5) | 33 (48.5) | | 33 (48.5) | 35 (51.5) | | 40 (58.8) | 28 (41.2) | |
| **Parity** | | | | | | | | | | | | | | | |
| Nulliparous | 391 (51.5) | 369 (48.6) | 0.360 | 378 (49.7) | 382 (50.3) | 0.982 | 398 (52.4) | 362 (47.6) | 0.165 | 404 (53.2) | 356 (46.4) | 0.027 | 371 (48.8) | 389 (51.2) | **0.080** |
| 2–4 | 427 (48.4) | 455 (51.6) | | 442 (50.1) | 440 (49.9) | | 422 (47.9) | 460 (52.2) | | 425 (48.2) | 457 (51.8) | | 443 (50.2) | 439 (49.8) | |
| ≥ 5 | 51 (53.7) | 44 (46.3) | | 48 (50.5) | 47 (49.5) | | 50 (52.6) | 45 (47.4) | | 39 (41.1) | 56 (59.0) | | 58 (61.1) | 37 (39.0) | |
| **Marital Status** | | | | | | | | | | | | | | | |
| Single | 39 (38.2) | 63 (61.8) | **0.014** | 46 (45.1) | 56 (54.9) | 0.305 | 55 (53.9) | 47 (46.1) | 0.418 | 46 (45.1) | 56 (54.9) | 0.305 | 57 (55.9) | 45 (44.1) | 0.223 |
| Married | 834 (50.8) | 809 (49.2) | | 827 (50.3) | 816 (49.7) | | 818 (49.8) | 825 (50.2) | | 827 (50.3) | 816 (49.7) | | 816 (49.7) | 827 (50.3) | |
| **Education** | | | | | | | | | | | | | | | |
| Primary or less | 25(51.0) | 24 (49.0) | **0.004** | 28 (57.1) | 21 (42.9) | 0.595 | 22 (44.9) | 27 (55.1) | 0.153 | 18 (36.7) | 31 (63.3) | **0.001** | 34 (69.4) | 15 (30.6) | **0.001** |
| Secondary | 221 (43.9) | 283 (56.2) | | 251 (49.8) | 253 (50.2) | | 236 (46.8) | 268 (53.2) | | 198 (39.3) | 306 (60.7) | | 285 (56.6) | 219 (43.5) | |
| Tertiary or more | 626 (52.7) | 562 (47.3) | | 591 (49.8) | 597 (50.3) | | 613 (51.6) | 575 (48.4) | | 654 (55.1) | 534 (45.0) | | 553 (46.6) | 635 (53.5) | |
| **Employment Status** | | | | | | | | | | | | | | | |
| Unemployed | 97(51.3) | 92 (48.7) | 0.706 | 106 (56.1) | 83 (43.9) | 0.078 | 106 (56.1) | 83 (43.9) | 0.078 | 107 (56.6) | 82 (43.4) | **0.055** | 97(51.3) | 92 (48.7) | 0.706 |
| Employed | 776 (49.9) | 780 (50.1) | | 767 (49.3) | 789 (50.7) | | 767 (49.3) | 789 (50.7) | | 766 (49.2) | 790 (50.8) | | 776 (49.9) | 780 (50.1) | |
| **Religion** | | | | | | | | | | | | | | | |
| Christianity | 525 (52.0) | 485 (48.0) | **0.067** | 521 (51.6) | 489 (48.4) | 0.149 | 483 (47.8) | 527 (52.2) | **0.032** | 561 (55.5) | 449 (44.5) | **0.001** | 468 (46.4) | 542 (53.7) | **0.001** |
| Islam | 345 (47.5) | 381 (52.5) | | 349 (48.1) | 377 (51.9) | | 385 (53.0) | 341 (47.0) | | 309 (42.6) | 417 (57.4) | | 399 (55.0) | 327 (45.0) | |
| **Ethnicity** | | | | | | | | | | | | | | | |
| Non-Yorubas | 97 (54.5) | 81 (45.5) | 0.206 | 94 (52.8) | 84 (47.2) | 0.429 | 55 (30.9) | 92 (69.1) | **0.000** | 116 (65.2) | 62 (34.8) | **0.001** | 83 (46.6) | 95 (53.4) | 0.326 |
| Yorubas | 774 (49.5) | 790 (50.5) | | 777 (49.7) | 787 (50.3) | | 816 (52.2) | 748 (47.8) | | 755 (48.3) | 809 (51.7) | | 790 (50.5) | 774 (49.5) | |
| **Income** | | | | | | | | | | | | | | | |
| <20,000 | 261 (44.8) | 322 (55.2) | **0.012** | 271 (46.5) | 312 (53.5) | 0.298 | 276 (47.3) | 307 (52.7) | 0.551 | 233 (40.0) | 350 (60.0) | **0.001** | 317 (54.4) | 266 (45.6) | **0.002** |
| 20,000–99,999 | 442 (52.4) | 401 (47.6) | | 427 (50.7) | 416 (49.4) | | 421 (49.9) | 422 (50.1) | | 450 (53.4) | 393 (46.6) | | 401 (47.6) | 442 (52.4) | |
| ≥ 100,000 | 58 (53.7) | 50 (46.3) | | 52 (48.2) | 56 (51.9) | | 57 (52.8) | 51 (47.2) | | 71 (65.7) | 37 (34.3) | | 41 (38.0) | 67 (62.0) | |

women with high education consumed more refrained grains than less educated women: (primary "29.3%" versus tertiary "41.7%"). The dietary pattern also differed by religion, as Muslims had a higher consumption of a protein-rich diet with non-alcoholic beverages (p = 0.001) and legumes (p = 0.001) but a lower intake of the typical diet with alcohol (p = 0.024). Christians consumed more refrained grains compared to Muslims (p = 0.001). Furthermore, those who earn <N20, 000 presented high consumption of a 'protein-rich diet with non-alcoholic beverages' than those earning more than N100 000; 322 (55.2%) vs 50 (46.3%). Similarly, legume consumption was higher among those who earn <N20, 000 than those who earn more than N100 000; 350 (60.0%) vs 37 (34.3%).

The modelling of the participants' sociodemographic characteristics and dietary patterns with the unadjusted and adjusted β and 95% CI are shown in Table 5. The intake of the typical diet with alcohol decreased with the woman's level of education in a monotonic fashion: secondary school; adjusted β:-0.69 (-1.35; -0.02) p = 0.043 and tertiary education; adjusted β:-0.83 (-1.49; -0.018) p = 0.013 compared with women with primary school education only. Conversely, women with tertiary education had a higher mean dietary score for refined grains [adjusted β: 0.50 (- 0.04; 0.95) p = 0.033] compared with women with primary school education only. Income had an inverse association with the consumption of a protein-rich diet with non-alcoholic beverages, a typical diet with alcohol, and legumes in a dose-response fashion.

Also, employed women had a higher mean of fruits [adjusted β: 0.33 (0.02; 0.65) p = 0.040] compared with women without employment. Parity had a significant association with the intake of a typical diet with an alcoholic beverage as multiparous women [adjusted β: 0.30 (0.08; 0.52) p = 0.009] had significantly higher consumption compared with nulliparous women. Muslims had a higher intake of legumes [adjusted β: 0.28 (- 0.92; 0.47) p = 0.004] and a protein-rich diet with non-alcoholic beverage [adjusted β: 0.25 (0.01; 0.50) p = 0.039] compared to Christians.

## Discussion

Maternal nutrition is an important modifiable factor for optimal foetal development, pregnancy and neonatal outcomes, and mitigating the future risk of non-communicable diseases among women of reproductive age [19–21]. Therefore, understanding the dietary patterns of pregnant women, especially in LMIC societies undergoing epidemiologic and nutritional transitions such as Nigeria, is essential for predicting disease risk, formulating nutritional policies and providing nutritional interventions for pregnant women. We identified five dietary patterns: "protein-rich diet with non-alcoholic beverage", "fruits", and "typical diet with alcohol" legumes and "refined grains" among pregnant women in this study. We also ascertained the sociodemographic factors associated with maternal dietary patterns in Ibadan, Nigeria. Dietary pattern using factor analysis is gaining some attention in Nigeria but has been scarcely examined among pregnant women. Nwaru et al. (2012) examined the dietary pattern through a 24-hour dietary recall among mothers and children using Nigerian Demographic and Health Surveys data [22]. Unlike the food frequency questionnaire used in this study, the 24-hour dietary recall does not capture habitual dietary patterns. Recently researchers have begun examining the dietary pattern of specific Nigerian sub-populations, including school children [23] out-of-school adolescents [24], university undergraduates [25], and households [26].

The protein-rich diet with non-alcoholic beverages explained the highest variance in the dietary pattern in our study population. This dietary pattern was essentially healthy and nutrient-dense because it provided multiple sources of macro and micronutrients from animal protein–fish, eggs and red meat–an essential nutrient for foetal growth and development. Green leafy vegetables are important sources of minerals and vitamins (vitamins A, C, K, and E,

**Table 5. Modelling the participants' sociodemographic characteristics and dietary patterns.**

| | Protein-rich diet and non-alcoholic beverages | | | | Fruits | | | | Typical diet with alcohol | |
|---|---|---|---|---|---|---|---|---|---|---|
| | Unadjusted β coefficient (95% CI) | p-value | Adjusted β coefficient (95% CI) | p-value | Unadjusted β coefficient (95% CI) | p-value | Adjusted β coefficient (95% CI) | p-value | Unadjusted β coefficient (95% CI) | p-value |
| **Age group** | | | | | | | | | | |
| < 20 | Ref | | Ref | | Ref | | Ref | | Ref | |
| 20–29 | -0.35 (-1.12; 0.43) | 0.383 | 0.49(-.49; 1.46) | 0.327 | -0.13 (-0.86; 0.61) | 0.737 | 0.40 (-0.40; 1.19) | 0.327 | 0.30 (-0.42; 1.02) | 0.417 |
| 30–39 | -0.47 (-1.25; 0.31) | 0.237 | 0.60(-.39; 1.58) | 0.234 | -0.07 (-0.81; 0.66) | 0.844 | 0.45 (-0.36; 1.26) | 0.279 | 0.31 (-0.41; 1.03) | 0.404 |
| ≥ 40 years | - 0.80 (-1.72; 0.13) | 0.093 | 0.61(-.51; 1.72) | 0.284 | -0.06(-0.94; 0.82) | 0.887 | 0.37 (-0.54; 1.29) | 0.423 | 0.17 (-1.18; 1.03) | 0.699 |
| **Parity** | | | | | | | | | | |
| Nulliparous | Ref | | Ref | | Ref | | Ref | | Ref | |
| ≥ 1 | 0.20 (-0.02; 0.41) | 0.070 | -0.08 (-0.31; 0.16) | 0.521 | -0.05 (-0.25; 0.15) | 0.614 | 0.12 (-.49; 0.40) | 0.216 | 0.31 (0.11; 0.50) | 0.002 |
| **Marital Status** | | | | | | | | | | |
| Single | Ref | | Ref | | Ref | | Ref | | Ref | |
| Married | -.0.26 (-0.71; 0.19) | 0.251 | -0.84 (-1.38; 0.30) | 0.002 | -0.38 (-0.80; - 0.04) | 0.078 | -.04 (-0.48; 0.40) | 0.847 | 0.07 (-0.35; 0.48) | 0.743 |
| **Education** | | | | | | | | | | |
| Primary or less | Ref | | Ref | | Ref | | Ref | | Ref | |
| Secondary | -0.19 (-0.81; 0.59) | 0.578 | -0.10 (-0.80; 0.60) | 0.773 | -0.07 (-0.69; 0.55) | 0.835 | -.33 (-0.88; 0.22) | 0.238 | -0.74 (-1.34; -0.13) | 0.017 |
| Tertiary | -0.63 (-1.27; 0.01) | 0.052 | -0.44 (-1.14; 0.25) | 0.206 | -0.14 (-0.75; 0.46) | 0.643 | -.77 (-1.32; -0.28) | 0.005 | -0.98 (-1.57; -0.39) | 0.001 |
| **Employment Status** | | | | | | | | | | |
| Unemployed | Ref | | | | | | | | | |
| Employed | 0.31 (-0.03; 0.65) | 0.072 | 0.18 (-0.49; 0.86) | 0.590 | 0.33 (0.02; 0.65) | 0.040 | 0.38 (-0.17; 0.93) | 0.178 | 0.44 (0.12; 0.75) | 0.006 |
| **Religion** | | | | | | | | | | |
| Christianity | Ref | | Ref | | Ref | | Ref | | Ref | |
| Islam | 0.40 (0.19; 0.62) | 0.000 | 0.25 (0.01; 0.50) | 0.039 | 0.16 (-0.05; 0.36) | 0.131 | 0.27 (-0.07; 0.47) | 0.007 | -0.01 (-0.21;- 0.19) | 0.900 |
| **Ethnicity** | | | | | | | | | | |
| Non-Yorubas | Ref | | Ref | | Ref | | Ref | | Ref | |
| Yorubas | 0.49 (0.15; 0.84) | 0.005 | -.62 (-.89; 0.12) | 0.188 | -0.03 (-0.36; 0.30) | 0.867 | 0.14 (-0.17; 0.44) | 0.386 | -0.49 (-0.81; -0.16) | 0.003 |
| **Income** | | | | | | | | | | |
| <20,000 | Ref | | Ref | | Ref | | Ref | | Ref | |
| 20,000–99,999 | -0.51(-0.74; - 0.27) | 0.001 | -0.37(-0.62; - 0.12) | 0.004 | -0.20(-0.42; 0.02) | 0.079 | -.42(-0.63; - 0.23) | <0.001 | -0.38 (-0.60; -0.15) | 0.001 |
| ≥ 100,000 | -0.60(-1.06; - 0.14) | 0.011 | -0.38(-0.85; 0.09) | 0.111 | -0.18(-0.61; -0.26) | 0.421 | -.51(-0.89; -0.14) | 0.007 | -0.53 (-0.97; -0.09) | 0.019 |

| | Typical Diet with alcohol | | Legumes | | | | Refined Cereals | | | |
|---|---|---|---|---|---|---|---|---|---|---|
| | Adjusted β coefficient (95% CI) | p-value | Unadjusted β coefficient (95% CI) | p-value | Adjusted β coefficient (95% CI) | p-value | Unadjusted β coefficient (95% CI) | p-value | Adjusted β coefficient (95% CI) | p-value |
| **Age group** | | | | | | | | | | |
| < 20 | Ref | | Ref | | Ref | | Ref | | | |
| 20–29 | 0.35 (-.67; 1.37) | 0.505 | 0.06 (-0.55; 0.68) | 0.842 | 0.32 (-0.66; 1.29) | 0.527 | -0.02 (-.56; 0.53) | 0.952 | | |
| 30–39 | 0.27 (-.76; 1.31) | 0.610 | 0.03 (-0.59; 0.64) | 0.937 | 0.18 (-0.81; - 1.17) | 0.727 | -0.08 (-0.63; 0.47) | 0.779 | | |
| ≥ 40 years | - 0.11 (-1.27; 1.06) | 0.860 | 0.07(-0.67; 0.81) | 0.849 | 0.01(-1.12; 1.13) | 0.992 | -0.05 (-071; 0.60) | 0.871 | | |
| **Parity** | | | | | | | | | | |
| Nulliparous | Ref | | Ref | | Ref | | Ref | | | |
| ≥ 1 | 0.30 (0.08; - 0.52) | 0.009 | 0.19 (0.02; 0.36) | 0.028 | 0.14 (-0.35; 0.32) | 0.115 | 0.02 (-0.13; 0.17) | 0.776 | | |
| **Marital Status** | | | | | | | | | | |
| Single | Ref | | Ref | | Ref | | Ref | | | |

*(Continued)*

**Table 5.** (Continued)

| | | | | | | | | | | |
|---|---|---|---|---|---|---|---|---|---|---|
| Married | -.0.32 (-0.89; 0.25) | 0.265 | -0.76 (-0.43; 0.28) | 0.676 | -.16 (-.71; 0.38) | 0.562 | 0.07 (-0.25; 0.38) | 0.679 | | |
| **Education** | | | | | | | | | | |
| Primary or less | Ref | | Ref | | Ref | | Ref | | Ref | |
| Secondary | -0.69 (-1.35; -0.02) | 0.043 | -0.52 (-1.03; -0.01) | 0.046 | -0.36 (-0.90; 0.18) | 0.194 | 0.29 (-0.18; 0.75) | 0.223 | 0.27 (-0.19; 0.73) | 0.253 |
| Tertiary | -0.83 (-1.49; -0.18) | 0.013 | -.1.11(-1.61; -0.61) | 0.001 | -0.78 (-1.32; -0.24) | 0.005 | 0.54 (0.09; 0.99) | 0.019 | 0.50 (-0.04; 0.95) | 0.033 |
| **Employment Status** | | | | | | | | | | |
| Unemployed | Ref | | | | | | Ref | | | |
| Employed | 0.27 (-0.40; 0.95) | 0.431 | -0.32(-0.54; -0.59) | 0.019 | 0.38 (-0.17; 0.993) | 0.174 | 0.07 (- 0.17; 0.31) | 0.578 | | 0.967 |
| **Religion** | | | | | | | | | | |
| Christianity | Ref | | Ref | | Ref | | Ref | | Ref | |
| Islam | 0.19 (-.06; 0.44) | 0.134 | 0.48 (0.31; 0.65) | 0.001 | 0.28(0.92–0.47) | 0.004 | - 0.18 (-.33; - 0.26) | 0.021 | -0.10 (-0.26; 0.05) | 0.196 |
| **Ethnicity** | | | | | | | | | | |
| Non-Yorubas | Ref | | Ref | | Ref | | Ref | | | |
| Yorubas | 0.31 (-0.08; 0.71) | 0.115 | 0.42 (0.15; 0.70) | 0.003 | -.59 (-.97; 0.12) | 0.002 | -0.18 (-0.42; -0.07) | 0.156 | | |
| **Income** | | | | | | | | | | |
| <20,000 | Ref | | Ref | | Ref | | Ref | | | |
| 20,000–99,999 | -0.32(-0.56; -0.08) | 0.009 | -0.59(-0.77; - 0.40) | 0.001 | -0.42(-0.62; 0.22) | 0.004 | 0.07(-0.10; -0.24) | 0.447 | | |
| ≥ 100,000 | -0.47(-0.93--0.02) | 0.065 | -0.74(-1.10; - 0.38) | 0.001 | -0.51(-0.88; - 0.14) | 0.376 | 0.05 (-0.28; 0.38) | 0.776 | | |

including calcium, iron, fibre and folate, which are essential for preventing neural tube defects. This pattern was also rich in milk, sugar-sweetened beverages (SSBs) and added sugars from pastries. Cream milk benefits women and the growing foetus because of the high calcium content required for solid bones and cell function.

Conversely, pregnant women should consume SSBs and added sugars in moderation or eliminate and replace them with low or no calorie-containing drinks such as water, particularly women that are obese [27]. SSBs have been associated with poor dietary quality [28], high energy intake [29], weight gain [30] and increased cardiometabolic risk that results from the spike of blood glucose and insulin levels, and high glycaemic load leading to decreased insulin sensitivity [31–34]. Additionally, the protein-rich diet with non-alcoholic beverages was inversely associated with income; women with lower income consumed a more protein-rich diet with non-alcoholic beverages than women with high income. This implies that these food items are likely inexpensive, readily accessible and available to women. An inverse association with socioeconomic status was also observed for the typical diet with alcoholic beverage and legumes dietary pattern.

The typical diet with alcohol consisted of commonly consumed food items in Nigeria, namely pando/pounded yam, fermented cassava puddling (*fufu*), pork, snail, oil-based soups, soya milk and importantly alcoholic drinks such as beer and palm wine. The alcoholic content makes it a harmful dietary pattern because of its teratogenic effects and the associated adverse pregnancy and developmental outcomes. For example, alcohol ingestion during pregnancy is the leading cause of preventable congenital anomalies in developed countries [35]. There is growing evidence of a rise in the intake of alcohol among women of reproductive age, especially in developing countries [36, 37]. In sub-Saharan Africa, the rise in alcohol consumption among women has been linked to urbanisation, economic growth, increasing social acceptability of the habit, changing gender roles and so on [37]. The WHO has stipulated that no amount

of alcohol is safe during pregnancy and that pregnant women should abstain from alcohol to prevent the associated adverse perinatal and developmental outcomes [38, 39]. Therefore, it is necessary to assess alcohol intake and encourage abstinence during antenatal care. This dietary pattern increased with parity, suggesting that women with higher parity than nulliparous women reported a higher intake of alcohol-containing diet. This association has been reported by other researchers in Africa [40, 41]. Additionally, socioeconomic status had a negative association with this dietary pattern, i.e., women with low education and income had a higher consumption of this alcohol-based diet, perhaps due to a lack of awareness of the adverse effects of alcohol consumption during pregnancy. A study in Uganda reported that the availability of cheap alcoholic drinks and their free distribution during celebrations make alcoholic intake common among low-income earning women [40]

The legumes and fruit patterns were homogenous groups explaining 4.1% and 2.8% variations, respectively. Legumes are plant proteins rich in dietary fibre with a low-glycaemic index (GI). Legumes are a healthy and inexpensive diet high in phytochemicals, fibre, proteins, minerals and vitamins [42]. Legumes enhance cardio-metabolic health by maintaining insulin sensitivity, improving lipid profiles, preventing insulin resistance [43], obesity [44], and cardiovascular risk scores [45] and these have been well reported in the literature. During pregnancy, legumes are beneficial in preventing gestational diabetes and excessive weight gain by maintaining postprandial glucose excursions, blood glucose and insulin levels. Legumes are, however, underutilised in our environment and often consumed by more impoverished individuals and families. Our study showed that legume consumption declined significantly in a dose-response fashion with the level of education and income. For example, the mean dietary score of legumes for women with tertiary education was much less [Adjusted β -.78 (p = 0.005)] compared with women with primary education or less. Some reasons for the underutilisation of legumes, especially among educated women, might be their prolonged cooking time and less palatability. It might also be associated with gastrointestinal side effects like increased flatulence, among others [46, 47]. For instance, legumes are regarded as the poor man's meat [46], and our study shows higher consumption among low-income women. Hence the need to encourage women to consume legumes in various forms and find innovative ways of preparing them.

The fruit pattern was clearly identified in this study population. Fruits and vegetables are rich in vitamins and minerals: A, C, E, magnesium, zinc, phosphorus, folic acid, fibre and antioxidants but low in calorie and dietary fat. The benefit of eating fruits and vegetables derived from their antioxidants, vitamins and phytochemicals [48]. Fruits were only significantly associated with women's employment status, with employed women having a higher mean intake of fruits than unemployed women [Adjusted β 0.33 p = 0.040)]. This implies a lack of access to fruits because of cost; hence, only employed women can readily access fruits.

The refined grains pattern was high in rice and refined cereals, which have a high GI and can increase the risk of metabolic dysfunctions because they are lower in fibre and essential nutrients than whole grains [49, 50]. In this study, high intakes of refined cereals were associated with higher education and income. Refined cereals are usually fortified with micronutrients and quick to prepare but are high in added sugars and expensive [51–53]. Hence we noted a higher level of consumption among women with higher education than those with primary education. Similarly, nulliparous women were also observed to have a high intake of refined grains, which may explain their susceptibility to obesity in subsequent pregnancies due to excessive weight gain and postpartum weight retention [54, 55].

This study is likely the first to describe the dietary pattern of pregnant women using a data-driven technique such as PCA in Nigeria. We also identified factors that can influence the dietary patterns of pregnant women in this population, which can inform public health

interventions, especially behavioural change communication during antenatal care. These findings are likely applicable to women across all spectrums of reproductive age and crucial for designing public health policies and advisories to guide public health interventions for women's health and quality of life in LMICs.

However, our study also has limitations. First, the dietary assessment was conducted using a qualitative food frequency questionnaire without quantifying portion sizes. Also, the dietary assessment was conducted at baseline; hence the study could not account for dietary changes during pregnancy. We relied on participants' recollection of food consumption, and bias in reporting healthy eating habits is not unlikely, particularly among women with higher education. The other limitations are those associated with the complex hierarchical nature of data-driven techniques like factor analysis.[9, 22].

## Conclusion

Prenatal nutrition impacts birth outcomes and is also an essential modifiable factor. We described five dietary patterns of pregnant women using a data-driven technique such as PCA in Nigeria. We also identified factors influencing maternal dietary patterns, which can inform public health policy and interventions, especially behavioural change communication during antenatal care.

## Acknowledgments

We would like to thank our research team for their dedication, support and hard work–research nurses, laboratory scientists, research assistants Data personnel. We also wish to appreciate the health workers–doctors, nurses, and clinic staff as well as the record staff of the various health facilities for their cooperation and support in the four facilities: University College Hospital, Adeoyo Maternity Teaching Hospital, Jericho Specialist Hospital, and Saint Mary Catholic Hospital Oluyoro, Ibadan. We appreciate the input of CARTA (Consortium for Advanced Research Training for Africa) for all its training, care, support, oversight, funding and sponsorship efforts. The contributions of Dr Fagbamigbe's input into the manuscript are acknowledged. The input of Prof Rasaki Sanusi and Dr Folake Samuel of the Department od Human Nutrition and Dietetics, University of Ibadan, in developing the food frequency questionnaire is appreciated.

## Author Contributions

**Conceptualization:** Ikeola A. Adeoye.

**Data curation:** Ikeola A. Adeoye.

**Formal analysis:** Ikeola A. Adeoye.

**Funding acquisition:** Ikeola A. Adeoye.

**Investigation:** Ikeola A. Adeoye.

**Methodology:** Ikeola A. Adeoye, Akinkunmi P. Okekunle.

**Project administration:** Ikeola A. Adeoye.

**Resources:** Ikeola A. Adeoye, Akinkunmi P. Okekunle.

**Software:** Ikeola A. Adeoye, Akinkunmi P. Okekunle.

**Supervision:** Ikeola A. Adeoye, Akinkunmi P. Okekunle.

**Validation:** Ikeola A. Adeoye, Akinkunmi P. Okekunle.

**Visualization:** Ikeola A. Adeoye, Akinkunmi P. Okekunle.

**Writing – original draft:** Ikeola A. Adeoye.

**Writing – review & editing:** Ikeola A. Adeoye, Akinkunmi P. Okekunle.

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
