## [Decision Letter · Decision Letter 0]

24 May 2022

PONE-D-22-06619Dietary patterns and associated factors among pregnant women in Ibadan, Nigeria: Evidence from Ibadan Pregnancy Cohort StudyPLOS ONE

Dear Dr. Adeoye,

Thank you for submitting your manuscript to PLOS ONE. After careful consideration, we feel that it has merit but does not fully meet PLOS ONE’s publication criteria as it currently stands. Therefore, we invite you to submit a revised version of the manuscript that addresses the points raised during the review process.

We look forward to receiving your revised manuscript.

Kind regards,

Linglin Xie

Academic Editor

PLOS ONE

Journal Requirements:

Reviewers' comments:

Reviewer's Responses to Questions

**Comments to the Author**

1. Is the manuscript technically sound, and do the data support the conclusions?

Reviewer #1: Yes

2. Has the statistical analysis been performed appropriately and rigorously? 

Reviewer #1: Yes

3. Have the authors made all data underlying the findings in their manuscript fully available?

Reviewer #1: Yes

4. Is the manuscript presented in an intelligible fashion and written in standard English?

Reviewer #1: No

5. Review Comments to the Author

Reviewer #1: As you have described, prenatal nutrition impacts birth outcomes and is also an important factor that can potentially be modified. Your research will be beneficial in informing public health education, interventions, and policy.

Overall, the paper is well written and succinctly describes your study methods and results. However minor editing of the paper is needed, as in some places words are missing or there are minor grammatical errors (including in lines 91, 103, 127 (unknown what the * references), 130, 158, 163, 211, 360, 372, 374, 40-408, 414, 416, 420, 427, 450).

Specific recommendations:

• Lines 167-168: Please provide another sentence or two describing how the dietary information was transformed and harmonized

• Lines 218-224: To provide clarity for the reader, these results need to have consistency in the format of reporting, particularly related to use of the ( ) and [ ] and mentioning daily and weekly, for example use a format like this in reporting all the results in this paragraph:

o [294 (45.5% daily, 898 (51.4% weekly]

• Page 18, table 5: Please define “Exotic Diet with alcohol” the term “Exotic Diet” is not described in the manuscript

• Lines 389-390: The study did not document that lower education and lower income caused higher alcohol consumption due to low health literacy regarding the adverse effects of alcohol, please redraft the sentence

• Line 441: another potential bias is response bias, as the respondents may not have been entirely truthful or may have only wanted to report “good” eating habits in responding to the interviewers’ questions

• Line 459-462: identify specific research/manuscript contributions of each individual author

6. PLOS authors have the option to publish the peer review history of their article (what does this mean?). If published, this will include your full peer review and any attached files.

Reviewer #1: No

---

## [Author Response · Author response to Decision Letter 0]

24 Jun 2022

POINT BY POINT RESPONSE TO THE REVIEWERS COMMENTS

5. Review Comments to the Author

Reviewer #1: As you have described, prenatal nutrition impacts birth outcomes and is also an important factor that can potentially be modified. Your research will be beneficial in informing public health education, interventions, and policy.

Thank you Sir

Overall, the paper is well written and succinctly describes your study methods and results. 

We thank the reviewer for the kind comment.

However minor editing of the paper is needed, as in some places words are missing or there are minor grammatical errors (including in lines 91, 103, 127 (unknown what the * references), 130, 158, 163, 211, 360, 372, 374, 40-408, 414, 416, 420, 427, 450).

We are grateful to the reviewer for succinctly pointing our attention to these grammatical errors. The entire manuscript has been revised to ensure clarity in expression and avoid grammatical errors

Specific recommendations:

• Lines 167-168: Please provide another sentence or two describing how the dietary information was transformed and harmonized

We have included the following statements in lines 207 - 237 of the revised manuscript to described how dietary information was harmonized in the revised manuscript. Please the statement below

“Details of the food and drink items in the FFQ and how they are classified into food groups are presented in Table 1. For each food or drink, participants reported the frequency of food consumption as follows: once daily, more than once daily (i.e. 2 -3 times daily): once weekly, more than once weekly (i.e. 2 -3 times weekly),): once monthly, more than once monthly (i.e. 2 -3 times monthly). The consumption frequency was harmonised into daily, weekly, monthly and rarely and transformed into the frequency of daily consumption. ” 

• Lines 218-224: To provide clarity for the reader, these results need to have consistency in the format of reporting, particularly related to use of the ( ) and [ ] and mentioning daily and weekly, for example use a format like this in reporting all the results in this paragraph:

o [294 (45.5% daily, 898 (51.4% weekly]

Edited

• Page 18, table 5: Please define “Exotic Diet with alcohol” the term “Exotic Diet” is not described in the manuscript

The correct description here is “Typical diet with alcohol”.

We have changed “Exotic Diet with alcohol” to “Typical diet with alcohol” in Table 5 of the revised manuscrip.

• Lines 389-390: The study did not document that lower education and lower income caused higher alcohol consumption due to low health literacy regarding the adverse effects of alcohol, please redraft the sentence

We have revised the sentence in lines 571 - 582 of the revised manuscript rephrases the discussion between alcohol intake and literacy.

Additionally, socioeconomic status had a negative association with this dietary pattern, i.e., women with low education and income had a higher consumption of this alcohol-based diet perhaps due to a lack of awareness of the adverse effects of alcohol consumption during pregnancy. A study in Uganda also reported that the availability of cheap alcoholic drinks and its free distribution during celebrations make alcoholic intake common among low income earning women (40)

• Line 441: another potential bias is response bias, as the respondents may not have been entirely truthful or may have only wanted to report “good” eating habits in responding to the interviewers’ questions

We have included the following sentence in line 704 -706 the revised manuscript to itemize this unique bias suggested by the reviewer. Thank you.

• Line 459-462: identify specific research/manuscript contributions of each individual author

We are grateful to the reviewer for this suggestion. We have included the following statements in lines 710 -713 of the revised manuscript to itemize the contribution of each individual author. Please see the statement below;

Author’s contributions

IAA designed and conducted the study and analyzed the data. IAA and APO interpreted the data and wrote the initial draft of the manuscript. IAA and APO reviewed and critically revised the manuscript. All authors read and approved the final manuscript.

---

## [Decision Letter · Decision Letter 1]

28 Jul 2022

PONE-D-22-06619R1Dietary patterns and associated factors among pregnant women in Ibadan, Nigeria: Evidence from Ibadan Pregnancy Cohort StudyPLOS ONE

Dear Dr. Adeoye,

Thank you for submitting your manuscript to PLOS ONE. After careful consideration, we feel that it has merit but does not fully meet PLOS ONE’s publication criteria as it currently stands. Therefore, we invite you to submit a revised version of the manuscript that addresses the points raised during the review process.

We look forward to receiving your revised manuscript.

Kind regards,

Linglin Xie

Academic Editor

PLOS ONE

Journal Requirements:

Reviewers' comments:

Reviewer's Responses to Questions

**Comments to the Author**

1. If the authors have adequately addressed your comments raised in a previous round of review and you feel that this manuscript is now acceptable for publication, you may indicate that here to bypass the “Comments to the Author” section, enter your conflict of interest statement in the “Confidential to Editor” section, and submit your "Accept" recommendation.

Reviewer #1: All comments have been addressed

2. Is the manuscript technically sound, and do the data support the conclusions?

Reviewer #1: Yes

3. Has the statistical analysis been performed appropriately and rigorously? 

Reviewer #1: Yes

4. Have the authors made all data underlying the findings in their manuscript fully available?

Reviewer #1: Yes

5. Is the manuscript presented in an intelligible fashion and written in standard English?

Reviewer #1: No

6. Review Comments to the Author

Reviewer #1: Thank you for the revised manuscript and addressing the suggested comments. A number of edits were made, however because of the revisions the manuscript will again benefit from minor English editing.

I didn't complete a detailed language review of the entire manuscript, but here are some initial edits I identified in the revised manuscript and there may be others:

Line #93, believe the correct word is "other" vs. "over"

Line #103, the verb should be "were" vs. "was"

Line #106, delete "that

Lines #138-139 and #141-143 seem to repeat the same information

7. PLOS authors have the option to publish the peer review history of their article (what does this mean?). If published, this will include your full peer review and any attached files.

Reviewer #1: No

---

## [Author Response · Author response to Decision Letter 1]

29 Jul 2022

POINT BY POINT RESPONSE TO THE REVIEWERS COMMENTS

6. Review Comments to the Author

Reviewer #1: Thank you for the revised manuscript and addressing the suggested comments. A number of edits were made, however because of the revisions the manuscript will again benefit from minor English editing.

I didn't complete a detailed language review of the entire manuscript, but here are some initial edits I identified in the revised manuscript and there may be others:

Line #93, believe the correct word is "other" vs. "over"

• Replaced with other (line 92 page 4)

Line #103, the verb should be "were" vs. "was"

• Replaced with were (line 101 page 4)

Line #106, delete "that (line 104 page 4)

• deleted

Lines #138-139 and #141-143 seem to repeat

---

## [Editor Report · Decision Letter 2]

16 Aug 2022

Dietary patterns and associated factors among pregnant women in Ibadan, Nigeria: Evidence from Ibadan Pregnancy Cohort Study

PONE-D-22-06619R2

Dear Dr. Adeoye,

We’re pleased to inform you that your manuscript has been judged scientifically suitable for publication and will be formally accepted for publication once it meets all outstanding technical requirements.

Kind regards,

Linglin Xie

Academic Editor

PLOS ONE
---

## [Editor Report · Acceptance letter]

5 Sep 2022

PONE-D-22-06619R2 

Dietary patterns and associated factors among pregnant women in Ibadan, Nigeria: Evidence from Ibadan Pregnancy Cohort Study 

Dear Dr. Adeoye:

I'm pleased to inform you that your manuscript has been deemed suitable for publication in PLOS ONE. Congratulations! Your manuscript is now with our production department. 

Kind regards, 

on behalf of

Dr. Linglin Xie 

Academic Editor

PLOS ONE